# Viral Immunogenicity Prediction by Machine Learning Methods

**DOI:** 10.3390/ijms25052949

**Published:** 2024-03-03

**Authors:** Nikolet Doneva, Ivan Dimitrov

**Affiliations:** Faculty of Pharmacy, Medical University-Sofia, 1000 Sofia, Bulgaria; ndoneva@pharmfac.mu-sofia.bg

**Keywords:** viral immunogens, in silico modelling, machine learning algorithms, immunogenicity prediction

## Abstract

Since viruses are one of the main causes of infectious illnesses, prophylaxis is essential for efficient disease control. Vaccines play a pivotal role in mitigating the transmission of various viral infections and fortifying our defenses against them. The initial step in modern vaccine design and development involves the identification of potential vaccine targets through computational techniques. Here, using datasets of 1588 known viral immunogens and 468 viral non-immunogens, we apply machine learning algorithms to develop models for the prediction of protective immunogens of viral origin. The datasets are split into training and test sets in a 4:1 ratio. The protein structures are encoded by E-descriptors and transformed into uniform vectors by the auto- and cross-covariance methods. The most relevant descriptors are selected by the gain/ratio technique. The models generated by Random Forest, Multilayer Perceptron, and XGBoost algorithms demonstrate superior predictive performance on the test sets, surpassing predictions made by VaxiJen 2.0—an established gold standard in viral immunogenicity prediction. The key attributes determining immunogenicity in viral proteins are specific fingerprints in hydrophobicity and steric properties.

## 1. Introduction

Immunogenicity is the ability of a foreign biomacromolecule (protein, lipid, carbohydrate, or a combination of them) to produce a humoral and/or cell-mediated immune response in the host organism. When the immune response triggered by a substance results in the creation of memory cells, that substance is classified as a protective immunogen. These protective immunogens, particularly those originating from pathogens, hold potential as vaccine candidates [1]. The critical and intensive first stage in vaccine design involves the identification of these protective immunogens. This process, which can be both time-consuming and expensive, finds significant aid from in silico methods, which can greatly reduce the time and cost of the subsequent experimental work [2].

Immunoinformatics is an interdisciplinary field that involves the use of computational tools and algorithms to study the immune system and its response to various pathogens and foreign substances, such as viruses and bacteria. Immunoinformatics is used to understand and predict aspects of immune responses, such as the interactions between antigens (substances that trigger an immune response) and antibodies, the identification of potential vaccine candidates, and the analysis of immune system components at a molecular level. It plays a crucial role in accelerating vaccine development, understanding autoimmune diseases, and advancing our knowledge of the immune system’s complexities.

The VaxiJen 2.0 webserver is a bioinformatic tool for in silico identification and prediction of protective antigens of different origins. To the best of our knowledge, it is the only tool for in silico identification and prediction of protective antigens of viral, bacterial, parasitic, fungal, and tumor origin [3,4]. VaxiJen 2.0 uses an alignment-independent method based on auto- and cross-covariance (ACC) transformation of protein sequences into uniform equal-length vectors and a partial least squares algorithm (PLS) for antigen prediction [3]. The amino acids in protein sequences are presented by z-descriptors, representing their main physicochemical properties. The VaxiJen 2.0 model derived for the prediction of viral antigenicity is trained on a set of 100 antigenic and 100 non-antigenic proteins [3]. The datasets of viral proteins were collected from publications published prior to 2006. A huge number of viral antigens have been discovered, and new algorithms for data analysis have been developed since the VaxiJen 2.0 server was released. Recently, our group has updated the model for the prediction of protective antigens of bacterial origin [5]. New models were derived on an extended dataset, and E-scale descriptors were used for the presentation of the physicochemical and structural properties of the amino acids in protein sequences.

In the present study, we applied a variety of machine learning (ML) algorithms to updated datasets of known viral immunogens and non-immunogens in order to improve the prediction performance of the VaxiJen server. The amino acids in the protein sequences were described by E-scale descriptors [6], and the resulting numerical strings were transformed into uniform vectors by auto- and cross-covariance (ACC) calculations [7]. The applied algorithms to a training set included Naive Bayse, Logistic regression, Support Vector Machine (SVM) with Sequential Minimal Optimization (SMO), instance-based k-nearest neighbor (iBK), Random Forest (RF), Extreme Gradient Boosting (XGBoost), and Multilayer Perceptron (MLP). The derived ML models were validated using an external test set and checked for robustness by Y-randomization. The best performing models were selected to be implemented in the new version 3.0 of the server VaxiJen. Additionally, we identified the most impactful attributes contributing to the immunogenicity of viral proteins.

## 2. Results

### 2.1. Dataset Characterization

Datasets of 1588 immunogenic and 468 non-immunogenic human viral proteins from 31 viral species (incl. different strains, isolates, and serotypes) were collected as described in Materials and Methods. The proteins were presented in files in Fasta format in the Appendix A. We studied the diversity of the proteins in the collected datasets by assessing their potential homology. The homology between the proteins was evaluated through an alignment of their sequences using local BLAST searches [8]. Specifically, each immunogen was compared to the remaining immunogens, each non-immunogen to the other non-immunogens, and each non-immunogen to all immunogens collectively. The degree of homology between two proteins was assessed by the BLAST bit-score determined after the sequence alignment of their sequences. Detailed information concerning every protein in the datasets and its homologous proteins was presented in the Appendix A. A summary of the results are presented in Table 1.

### 2.2. Data Preprocessing

The proteins in the collected datasets were presented as numerical strings of 5n E-scale descriptors, where *n* is the number of amino acid residues. The strings of different lengths were transformed into uniform vectors by ACC transformation with a lag of five. The lag of five equals the shortest peptide in the dataset. The dataset (positive and negative set) with uniform equal-length vectors of 125 variables was divided into a training set and a test set in a 4:1 ratio. The training set, which contained 1270 immunogenic and 374 non-immunogenic proteins, was used to derive ML models. The test set containing 318 immunogenic and 94 non-immunogenic proteins was used for external validation of the derived ML models. The flowchart for data preprocessing is presented in Figure 1.

The summary results from the sequence alignment showed that 393 of 412 (95.4%) proteins in the test set had their homologues in the training set.

### 2.3. ML Modeling and Validation

Several supervised ML algorithms were applied to the training set to derive classification models for immunogenicity prediction. The performance of the derived models was assessed by 10-fold cross-validation on the training set using accuracy, sensitivity, specificity, Matthews correlation coefficient (MCC), and Area under the ROC curve (AROC) as metrics. The data are shown in Table 2.

Based on overall performance, we selected MLP, XGBoost, and RF as the most promising algorithms for immunogenicity prediction of viral proteins. Next, we optimized the number of variables using the gain/ratio attribute selection classifier in the WEKA software tool version 3.8 [9]. The ranking method reduced the number of attributes from 125 to 108. Whereby the algorithms with an optimized number of variables were applied to the training set and validated on the test set. The performance of the selected algorithms after optimization is shown in Table 3.

On the same test set, VaxiJen 2.0 achieved a sensitivity of 0.833, specificity of 1.000, and an MCC of 0.738. Expectedly, the specificity reached 1.000 considering the negative set comprised proteins from viral proteomes predicted as non-immunogens by VaxiJen 2.0.

### 2.4. ML Models Robustness Assessment

We employed Y-randomization [10] to assess the risk of chance correlation in obtaining classification models. For evaluating the robustness of our best-performing models, we utilized MCC and AROC as metrics. Following Y-randomization, the MCC values for all models were exceedingly close to zero: −0.042 for the RF model, 0.003 for the XGBoost model, and −0.037 for the MLP model. Additionally, the AROC values for all three models were below 0.5:0.472 for the RF model, 0.484 for the XGBoost model, and 0.482 for the MLP model.

### 2.5. Analysis of Model Prediction on the Test Set

The prediction of every protein presented in the test set by the best-performing models was presented in the Appendix A, together with the number of homologues among immunogens and non-immunogens of this protein. The summary of the results is presented in Table 4.

The majority voting of the best-performing models correctly determined 313 of 318 immunogens as protective antigens (TP) and 83 of 94 non-immunogens as non-antigens (TN). These results enable the calculation of classification metrics for the majority voting of the best-performing models: 0.966 for sensitivity, 0.943 for specificity, 0.954 for accuracy, and 0.888 for MCC.

### 2.6. Attribute Importance Assessment

The attribute importance for the chosen ML models was assessed using the ClassifierAttributeEval method within the WEKA software. Figure 2 displays the top 10% of attributes, ordered by their importance, for each model. Attributes common to all models are highlighted in bold, while unique attributes specific to a particular model are indicated by being underlined.

## 3. Discussion

The initial step in constructing in silico models involves gathering and preprocessing data. Data sources for viral protein immunogenicity encompass scientific literature, clinical trials, and databases. However, only a fraction pertains to human studies. We meticulously curated a dataset of 1588 immunogenic viral proteins by exhaustively screening PubMed (https://pubmed.ncbi.nlm.nih.gov/, accessed on 1 January 2021) and the Immune Epitope Database (https://www.iedb.org/, accessed on 1 June 2021) for relevant papers on viral immunogens in human studies. Subsequently, we searched for their sequences in UniProt (https://www.uniprot.org/, accessed on 14 June 2021) and NCBI (https://www.ncbi.nlm.nih.gov/, accessed on 1 June 2021) databases. Additional viral antigen studies in human trials were accessed via the clinical trials database (https://clinicaltrials.gov/, accessed on 1 July 2021).

Assessing the diversity within a set of proteins commonly involves evaluating their homology, wherein two proteins are considered homologous if their sequences exhibit substantial similarity. Sequence similarity searching, often implemented using the BLAST algorithm, stands out as the most prevalent and dependable method for characterizing sequences. The bit-score emerges as a dependable metric for deducing homology, with a significance threshold typically set at 40 for average-length proteins in searches conducted within protein databases containing fewer than 7000 entries [11]. The overview of protein homology data revealed that the majority of immunogens (1566) exhibited homology with other immunogens, with only 22 (1.4%) lacking immunogen homologues. The number of homologues for immunogens was distributed relatively evenly, ranging from 1 to the maximum value (194). In contrast, 368 (78.6%) of non-immunogens did not share homologues with immunogens, while 377 (80.5%) had homologues among other non-immunogens. These findings suggest a well-curated dataset where immunogens can be effectively distinguished from non-immunogens based on their primary sequence.

Building classification models necessitates both positive and negative sets. However, data on viral non-immunogens from human studies are scarce. To form the negative set for deriving classification models, we uploaded proteins from the positive set’s viral proteomes to the VaxiJen 2.0 webserver and selected proteins predicted as non-immunogens. Due to the limited proteins in viral proteomes, our negative set comprised only 468 non-immunogens, significantly smaller than the positive set. Yet, the final dataset of 1588 viral immunogens and 468 non-immunogens, while imbalanced, notably surpasses the dataset size used in the VaxiJen 2.0 model (100 immunogens and 100 non-immunogens). The superior number of immunogens and the high degree of homology among them explained the abundance of proteins in the test set that shared similarities with those in the training set.

Modeling protein biological function in silico encounters three key challenges: selecting suitable descriptors for protein sequences, comparing proteins of different lengths, and choosing algorithms for model derivation. To represent protein structures as numerical vectors, we utilized E-scale descriptors associated with amino acid physicochemical properties in protein primary structures, such as hydrophobicity, molecular size, steric properties, partial specific volume, relative frequency in sequences, and propensity for α-helix and β-strand formation.

To address the length variation challenge, we employed an auto- and cross-covariance (ACC) transformation. This transformation converted numerical strings of different lengths into uniform vectors. The lag variable in ACC facilitates considering adjacent amino acid influences. Auto-covariance captures linear relationships between identical E-descriptors of adjacent amino acids at different distances within the protein sequence, while cross-covariance deals with non-identical E-descriptors.

For model selection, we utilized classification metrics based on receiver operating characteristic (ROC) analysis. Given the imbalanced dataset, the Matthews correlation coefficient (MCC) and area under the ROC curve (AROC) were employed, aligning with best practices recommended by the US FDA-led project MAQC-II [12].

During 10-fold cross-validation on the training set (Table 2), the ML models exhibited varying performance in predicting immunogens and non-immunogens. Notably, Random Forest, SVM, and XGBoost excelled in predicting immunogens, while Logistic Regression, MLP, and XGBoost demonstrated superior accuracy and specificity in identifying non-immunogens. Ultimately, MLP, XGBoost, and Random Forest, which displayed the highest AROC values, were selected as the top-performing models.

Applying the Weka gain/ratio attribute selection algorithm optimized the attribute count from 125 to 108. Subsequently, RF and MLP models derived from the optimized training set outperformed their initial counterparts in 10-fold cross-validation (Table 2). However, the XGBoost model from the optimized set did not exhibit improved metrics and, in some cases, showed a slight decline. All three ML models demonstrated comparable or improved performance on the external test set compared to 10-fold cross-validation. This trend possibly reflects the imbalanced training set and indicates robust predictive capabilities without overestimation (Table 3). Evaluation of models on the external test set yielded consistent results for MCC and AROC, signifying high-quality model performance.

The MLP model showcased superior accuracy (0.938) owing to its enhanced specificity in recognizing non-immunogens compared to the XGBoost and RF models. However, XGBoost and RF models exhibited superior sensitivity in identifying immunogens. Overall, the models demonstrated a higher ability to recognize immunogens than non-immunogens. Y-randomization tests for XGBoost, RF, and MLP models, using MCC and AROC as performance metrics, confirmed the models’ robustness, ruling out chance correlations.

Comparing the derived ML models’ performance with VaxiJen 2.0 on the external test set in terms of sensitivity and MCC revealed superior predictive ability for all derived models. The sensitivity of derived models ranged from 0.962 to 0.987, surpassing VaxiJen 2.0’s sensitivity of 0.833, and the MCC scores for derived models (0.865–0.866) outperformed VaxiJen 2.0 (0.738).

The examination of model predictions on the test set (Table 4) revealed that, for 7 out of 11 false positives (63.6%), the count of immunogen homologues exceeded that of non-immunogen homologues by a significant margin. However, the data for false negatives (FN) did not facilitate the exploration of a correlation between the number of FN and the count of immunogen and non-immunogen homologues, either due to their relative balance or the small number of false negatives.

The outstanding performance metrics derived from predictions on the proteins in the test set using majority voting from the top-performing models indicate that these models are suitable for implementation in the upcoming version of the VaxiJen server.

An analysis of the top 10% most important attributes for each model revealed five common attributes (Figure 2). Notably, ACC223, ACC121, and ACC112 were consistently ranked as the top three attributes across all models. ACC223 and ACC112 represent auto-covariance of E-descriptors of amino acids at specific distances in the protein sequence, capturing essential relationships between molecular size and steric properties, as well as hydrophobicity, respectively. ACC121, a cross-covariance between E1 and E2 descriptors, measures the correlation between hydrophobicity and the molecular size/steric properties of adjacent amino acids. Two additional attributes, ACC554 and ACC145, appeared in the top 10% for RF and XGBoost. ACC554 represents the autocovariance of E5 descriptors at lag 4, capturing β-strand formation propensities at specific intervals in the protein sequence. ACC145, a cross-covariance between E1 and E4 descriptors at lag 5, relates hydrophobicity with the partial specific volume at defined intervals in the protein sequence.

## 4. Materials and Methods

### 4.1. Dataset

#### 4.1.1. Dataset of Immunogenic Viral Proteins (Positive Set)

A dataset of viral immunogens acting as protective antigens in humans was created, based on exhaustive literature searches and validated sources such as PubMed, UniProt, NCBI, IEDB, and ClinicalTrials.gov accessed on 1 July 2021. The literature search was limited to viral proteins that have been shown to be immunogenic in humans.

Keywords: (virus OR viral) AND (candidate OR candidates OR subunit) AND (protects OR protect OR protection OR protective) AND (vaccine OR vaccines) AND (human).

Results by year: 2010–2021;Text availability: all;Article types: all.

To collect a positive set, the immunogenic viral sequences underwent additional preprocessing, involving the removal of proteins sharing identical ID numbers, those with duplicate sequences, and those containing unidentified amino acids. Consequently, the positive set comprises 1588 distinct immunogenic viral proteins derived from 31 viruses, recognized as protective antigens in humans.

#### 4.1.2. Dataset of Non-Immunogenic Viral Proteins (Negative Set)

During our comprehensive literature review focusing on viral proteins, we encountered a notable absence of data concerning non-immunogens within existing human studies. To curate the negative set of non-immunogenic proteins, we assessed the immunogenicity of proteins derived from the proteomes of each virus present in the positive dataset using VaxiJen 2.0. Proteins recognized as known immunogens were excluded from the viral proteomes. The remaining proteins underwent evaluation via VaxiJen 2.0, and only those predicted as non-immunogens were selected. This methodology culminated in a dataset comprising 468 non-immunogenic proteins.

#### 4.1.3. Training and Test Sets

The dataset, comprising viral immunogens and non-immunogens, was partitioned into training and test sets in a 4:1 ratio. Specifically, the training set comprised 1270 immunogenic and 374 non-immunogenic proteins, while the test set comprised 318 immunogenic and 94 non-immunogenic proteins. The training set was used for model development, while the test set was utilized for external validation of the derived models.

#### 4.1.4. Analysis of the Diversity of Viral Proteins in Datasets

The degree of homology between the viral proteins presented in the datasets was analyzed. We made a local BLAST search to align the sequences of each immunogen with other immunogens, each non-immunogen with other non-immunogens, and each non-immunogen with all the immunogens. The outcomes of the sequence alignment are detailed in tables within the Appendix A, provided in Excel files. Only the bit-score values surpassing 40 are included in these tables.

### 4.2. E-Scales Descriptors

In this study, we utilized E-scales descriptors to provide quantitative characterizations of viral protein sequences as numeric sequences. Originally introduced by Venkatarajan and Braun [6], these descriptors offer a quantitative description of the 237 physicochemical properties associated with the 20 naturally occurring amino acids found in proteins. They generated five distinct numerical values for each of these amino acids through multidimensional scaling. The first component, E1, strongly correlates with amino acid hydrophobicity. E2 provides insights into molecular size and steric properties. Components E3 and E5 delineate amino acid propensities for α-helix and β-strand formations, respectively. Lastly, component E4 accounts for partial specific volume, the number of codons, and the relative frequency of amino acids in proteins.

Consequently, each protein in our dataset was represented as a string of 5n elements, where n denotes the protein’s length. This transformation resulted in numeric sequence strings of varying lengths derived from the primary sequences of proteins.

### 4.3. Auto-Cross Covariance (ACC) Transformation

Once the primary structures of proteins were encoded into numeric sequences by E-scales descriptors, they were transformed into uniform vectors by auto- and cross-covariance (ACC) calculations. The ACC transformation of protein sequences was introduced in 1993 by Wold et al. [7] as an alignment-independent preprocessing method for converting the different-length polypeptide chains into uniform equal-length vectors.

Auto-covariance measures the linear dependency between observations of one E-descriptor at different lags. It helps identify the presence of patterns or relationships within the E-descriptors themselves. The auto-covariance calculates the sum of the products of two identical E-scales descriptors (*E_j_* and *E_j_*) for pairs of amino acids in the sequence, divided by the difference between the total number of amino acids in the peptide (*n*) and the lag value:(1)$${ACC}_{jj(lag)}=\sum_{i}^{n-lag}\frac{E_{j,i}\times E_{j,i+lag}}{n-lag}$$

*j*—E-scale (1–5)*n*—the number of amino acids in a sequence*i*—the amino acid position index (1, 2, 3, … n)*l*—*lag* (1, 2, … L)*E*—a value of the E-descriptor

Cross-covariance measures the linear dependency between two different E-scales descriptors at different lags. It helps identify the relationship or association between different E-scales descriptors. The cross-covariance is the sum of the products of two different E-scales descriptors (*E_j_* and *E_k_*) for pairs of amino acids in the sequence, divided by the difference between the total number of amino acids in the peptide and the lag value:(2)$${ACC}_{jk(lag)}=\sum_{i}^{n-lag}\frac{E_{j,i}\times E_{k,i+lag}}{n-lag}$$

*k*—E-scale (1–5)The rest of the variables are the same as the ones in Equation (1)

*Lag* is the length of the frame of contiguous amino acids, for which *ACC_jj_* and *ACC_jk_* are calculated. In our study, we selected a lag value of 5, corresponding to the length of the shortest peptide within our dataset. This lag value was applied consistently across our analysis.

### 4.4. Machine Learning Methods

The ML algorithms applied in the present study are described below. The WEKA software tool was used for data mining and the building of the ML models [9]. The ACC-transformed datasets were used as input in the models. The output from each model is 1 for the predicted immunogen or 0 for the predicted non-immunogen.

#### 4.4.1. Naïve Bayes

The Naïve Bayes machine learning algorithm is a probabilistic classification technique based on Bayes’ theorem with a “naïve” assumption of independence between features. The algorithm calculates the probability of a given protein belonging to a particular class (immunogen or non-immunogen) by considering the probabilities of each feature occurring in that class.

#### 4.4.2. Instance-Based K-Nearest Neighbor (iBK)

k-nearest neighbor (kNN) measures the distances between the test data and each of the training data and classifies a data point based on how its k neighbors are classified [13]. We used the default WEKA parameters for the kNN algorithm, with distance weighting equal to 1/distance and k = 1.

#### 4.4.3. Support Vector Machine (SVM) with Sequential Minimal Optimization (SMO)

Support Vector Machine (SVM) with Sequential Minimal Optimization (SMO) is a powerful machine learning algorithm used for classification tasks. SVMs are particularly effective in cases where data is not linearly separable in the input space by transforming it into a higher-dimensional feature space. SVM uses vectors (cases) to define a hyperplane between two classes of data by maximizing the margin between the two classes. The vectors (cases) that define the hyperplane are called support vectors [14]. SMO is an algorithm specifically designed for training SVMs using the Weka software tool. It is used to efficiently solve the optimization problem involved in finding the optimal hyperplane. SMO enhances the efficiency of training SVMs by iteratively optimizing the necessary parameters [15]. In the current study, we have used gamma = 100, cost = 1, and default parameters for WEKA SMO classification.

#### 4.4.4. Random Forest

Type of decision tree algorithm was used, which creates multiple decision trees and combines their results to produce a final prediction [16]. Each decision tree is built using a random subset of the data and a random subset of the features, which helps to reduce overfitting and improve the model’s performance. The final prediction is made by combining the outputs of all the decision trees by taking a majority vote (for classification problems). The RF algorithm was applied in the present study with default WEKA parameters, and the number of iterations was set up to 1000.

#### 4.4.5. Logistic Classifier

The logistic classifier uses the principle of logistic regression, which models the probability of belonging to a given class depending on a linear combination of the input variables. This combination is fed by a logistic function that returns between 0 and 1. If the probability is closer to 1, membership in the first class is predicted, and if it is closer to 0, membership in the second class is predicted. An optimization method is used to train the logistic classifier, which adjusts the model parameters so as to minimize the error between the predicted probabilities and the actual classes of the objects in the training set [17]. In our study, we used the WEKA Logistic classifier with the default values of parameters.

#### 4.4.6. Multilayer Perceptron (MLP)

The multilayer perceptron is a type of artificial neural network that consists of multiple layers of interconnected nodes, or neurons, which are arranged in a feedforward manner. The input layer receives the inputs, and the output layer produces the output predictions. The hidden layers, which are located between the input and output layers, perform intermediate computations. In our study, we used the WEKA MLP classifier with one hidden layer with 56 nodes, and the default values of parameters.

#### 4.4.7. Extreme Gradient Boosting (XGBoost)

Gradient boosting is a decision-tree-based ensemble ML algorithm proposed by Breiman [18] and later developed by other researchers [19]. XGBoost combines multiple weak models to create a stronger one. It uses a gradient-boosting algorithm that iteratively adds decision trees to the model, with each subsequent tree attempting to correct the errors made by the previous trees. The algorithm optimizes a loss function by adjusting the weights of the instances and features and by choosing the split points that minimize the loss. Here, we used maxdepth = 3, eta = 1, nthread = 7, nrounds = 250, and the default parameters of the WEKA wrapper for the XGBoost library in the R programming language version 4.1.3.

### 4.5. Classification Metrics to Evaluate the Performance of a Machine Learning Model

Sensitivity, specificity, accuracy, the Matthews correlation coefficient (MCC), and the Area under the receiver operating characteristic (ROC) curve are some of the main parameters used to evaluate the performance of a machine learning model.

Four outcomes are possible in ROC analysis: true positives (TP, true immunogen predicted as immunogen); true negatives (TN, true non-immunogen predicted as non-immunogen); false positives (FP, true non-immunogen predicted as immunogen); and false negatives (FN, true immunogen predicted as non-immunogen). On the basis of these outcomes, five classification metrics were calculated:Sensitivity (recall): The proportion of actual immunogens that are correctly classified as immunogens by the model (TP/total positives).Specificity: The proportion of actual non-immunogens that are correctly classified as non-immunogens by the model (TN/total negatives).Accuracy: The proportion of proteins correctly classified as immunogens or non-immunogens among all proteins in the dataset (TP + TN)/total).Area under the ROC curve (AROC) compares the trade-off between the true positive rate (sensitivity) and the false positive rate (1—specificity). AROC is a quantitative measure of predictive ability and varies from 0.5 for random prediction to 1.0 for perfect prediction.Matthews correlation coefficient (MCC) accounts for the quality of two-class classifications [20]. It produces a high score only if the prediction obtains good results in all four confusion matrix categories (true positives, false negatives, true negatives, and false positives), proportionally both to the size of positive elements and the size of negative elements in the dataset [21,22]. MCC accepts values in the interval [−1, 1], where 1 denotes perfect agreement, −1 full disagreement, and 0 indicates no correlation between the prediction and the ground truth [23].

### 4.6. Attribute Selection Classifier Algorithm by Weka

Once we had selected the algorithms with the highest performance based on the classification metrics evaluation, we optimized the number of variables, selecting the most significant and using the attribute selection classifier algorithm in Weka. Whereby the algorithms with an optimized number of variables from 125 to 108 were applied to the training set and validated on the test set.

The gain/ratio algorithm was used for attribute selection. Gain/ratio is a feature selection method to determine the best attribute to split the data. Information gain is a measure of the reduction in entropy or uncertainty provided by a particular attribute. It evaluates how well an attribute separates the data into classes. High information gain indicates that the attribute is good for splitting the data. Intrinsic information measures the potential information gain of an attribute, normalized by the split information. It is used to correct the bias of information gain towards attributes with many values. GainRatio is defined as the ratio of information gain to intrinsic information. It is used to avoid bias towards attributes with many values. Weka uses GainRatio as a criterion for evaluating the worth of an attribute. A higher GainRatio indicates a better attribute for splitting the data. The algorithm consists of the following steps:Calculate the information gain for each attribute based on the target variable.Calculate the intrinsic information for each attribute, which measures the potential information provided by the attribute itself.Calculate the gain ratio for each attribute by dividing the information gain by the intrinsic information.Select the attribute with the highest gain ratio as the splitting criterion.

### 4.7. Models’ Analysis

#### 4.7.1. Y-Randomization

Y-randomization (Y-scrambling, Y-permutation) is an approach for checking the robustness of the derived models [24]. This method allows testing whether the predictions made by the model are not made just by chance. The performance of the derived models was compared to that of models built for permuted (randomly shuffled) Y responses, based on the same attributes and algorithm. In our case, the scrambled response is the predicted class for the proteins in the dataset: immunogen or non-immunogen.

The Y response was randomly shuffled 10 times, and then a new model was derived using the same attributes (ACC values) and algorithm as the original model derived by the WEKA software. For the comparison of the performance of the originally derived models and the models derived after Y-randomization, we used the MCC and AROC.

#### 4.7.2. Attribute Importance

We assessed the attribute importance for each of the best-performing models: Random Forest, XGBoost, and Multilayer Perceptron, using the ClassifierAttributeEval built-in method of the WEKA software. It allows you to select the desired classification algorithm and ranks the attributes by their importance regarding the selected algorithm.

## 5. Conclusions

In this study, we developed several machine learning models using a dataset comprising 1588 distinct immunogenic viral proteins sourced from 31 viruses recognized as protective antigens in humans, alongside 468 non-immunogenic proteins from human viruses. The best-performing models based on XGBoost, Random Forest, and Multilayer Perceptron algorithms showcased robustness and reliability, surpassing the capabilities of the existing VaxiJen 2.0 webserver. An attribute analysis conducted on the training set highlighted the critical importance of molecular size and steric properties of amino acids separated by three positions, as well as the hydrophobicity of amino acids separated by two positions within the protein sequence, in predicting the immunogenicity of viral proteins.

## Figures and Tables

**Figure 1 ijms-25-02949-f001:**
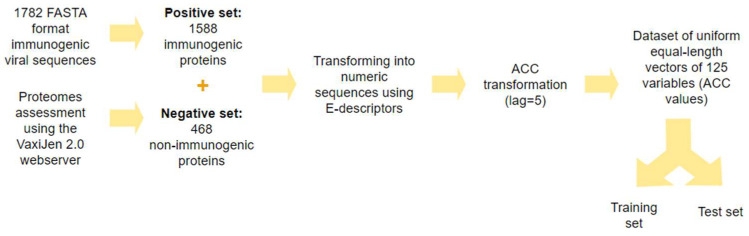
Flowchart of data preprocessing.

**Figure 2 ijms-25-02949-f002:**
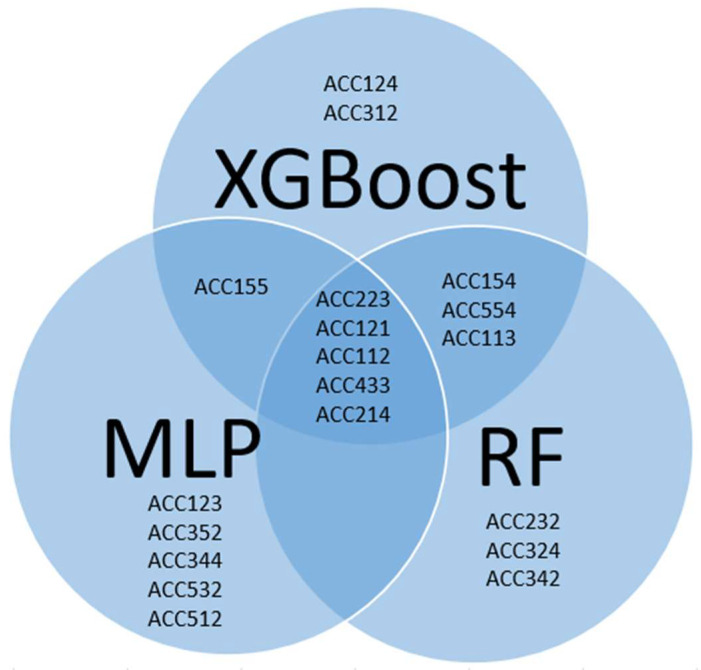
Top 10% of attributes ranked according to their importance for XGBoost, RF, and MLP models.

**Table 1 ijms-25-02949-t001:** Summary data from the sequence alignment results.

Proteins Data	Immunogens vs. Immunogens	Non-Immunogens vs. Non-Immunogens	Non-Immunogens vs. Immunogens
Number of proteins with a lack of homologues	22	91	368
Number of proteins with the highest number of homologues *	89 (194)	29 (45)	1 (194)
Number of proteins with one homologue	27	108	6
Number of proteins to align	1588	468	468

* The highest number of homologues is given in parenthesis.

**Table 2 ijms-25-02949-t002:** Performance of the machine learning (ML) models after 10-fold cross-validation on the training set.

Algorithm	Accuracy	Sensitivity	Specificity	MCC	AROC
Naive Bayes	0.662	0.837	0.487	0.319	0.790
Logistic classifier	0.865	0.957	0.816	0.784	0.925
Multilayer perceptron (MLP)	0.895	0.974	0.816	0.819	0.935
XGBoost	0.899	0.876	0.821	0.828	0.976
Support Vector Machine (SVM)	0.835	0.981	0.690	0.746	0.835
*k*-nearest neighbors (iBK)	0.857	0.979	0.735	0.773	0.871
Random Forest (RF)	0.872	0.986	0.757	0.805	0.976

**Table 3 ijms-25-02949-t003:** Performance of the best-performing machine learning (ML) models with an optimized number of variables on the training set after 10-fold cross-validation and the external test set.

Algorithm	Accuracy		Sensitivity		Specificity		MCC		AROC	
	Test	Training	Test	Training	Test	Training	Test	Training	Test	Training
XGBoost	0.917	0.896	0.984	0.972	0.851	0.821	0.866	0.819	0.977	0.973
RF	0.913	0.883	0.987	0.984	0.840	0.783	0.866	0.820	0.980	0.974
MLP	0.938	0.903	0.962	0.974	0.915	0.832	0.865	0.830	0.966	0.936

**Table 4 ijms-25-02949-t004:** FP and FN predicted proteins in the test set by majority voting (two of three) of the best performing models.

Protein ID	MLP	RF	XGBoost	Immunogen Homologues	Non-Immunogen Homologues
immunogens					
P13202	non-antigen	non-antigen	non-antigen	2	2
F5HCP3	non-antigen	non-antigen	non-antigen	0	2
P03169	non-antigen	non-antigen	non-antigen	2	2
Q07849	non-antigen	antigen	non-antigen	64	2
P06927	non-antigen	non-antigen	antigen	2	0
non-immunogens					
Q9QBF1	non-antigen	antigen	antigen	5	0
Q89745	antigen	antigen	antigen	17	0
A0A023LUC5	antigen	antigen	antigen	83	5
A0A059T563	antigen	antigen	non-antigen	83	5
H6QM96	non-antigen	antigen	antigen	86	3
X2DXM6	antigen	antigen	antigen	86	3
Q5V919	antigen	antigen	antigen	149	1
Q03963	antigen	antigen	antigen	0	5
P21044	antigen	antigen	antigen	0	5
Q77TG2	antigen	antigen	antigen	0	0
Q77TG4	antigen	antigen	antigen	0	0

## Data Availability

The models developed in the present study will be implemented in the updated version of VaxiJen 3.0.

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
