# Peer review of "Viral Immunogenicity Prediction by Machine Learning Methods"

_ijms, 2024, doi:10.3390/ijms25052949_

Round 1

Reviewer 1 Report

Comments and Suggestions for Authors

The development of computational methods of immunoinformatics is undoubtedly a very relevant and useful task, so this work is of interest not only for specialists working in silico, but also for experimental biologists, especially immunologists and virologists. In this regard, I would like to point out important shortcomings that should be corrected before publication.

Experimental biologists, if they want to publish their work, must provide the primary data on which their work was based (blots, immunohistochemistry results, in the case of proteomics, these are voluminous arrays of primary data). I think this rule should also apply to in silico work.

Authors used a relatively very small sample of about 2000 proteins, so why not provide data on these proteins as the supplementary information. I would like to see lists (exel tables, for example) of selected immunogens and non-immunogens, Which contain the names of the proteins, databases and/or articles where the proteins were taken from, UniProt or NCBI ID, names of viruses - sources of the proteins and their taxonomies. Additional data—on the functional types of the proteins (envelope protein, surface glycoprotein, or nucleoprotein) and its size—would also be helpful.

In the text of the article I would like to see a brief analysis of these data, including information about which viruses and which functional types of proteins were included in the study, and what degree of homology the proteins have with each other. Such an analysis will allow to assess the degree of diversity of the studied protein array.

The second proposal concerns the results of the study, namely the results of the analysis of the training set using ML algorithms. The training set is small, and in this regard, I would like to see exactly which proteins got there, and how they were distributed into two groups (immunogens and non-immunogens) by different ML algorithms. Are there correlations with functional classes or with the level of homology between proteins?

MINOR

1. For better visualization of the rezults, when comparing the performances of different algorithms, Venn diagrams could be used in addition to tables.

2. When choosing keywords for a literature search, a very strange choice (protects OR protect OR protection OR protective) instead of, for example, the much more logical choice (antigen, immunogen, epitope and their derivatives).

Comments on the Quality of English Language

Minor editing of English language required

Reviewer 2 Report

Comments and Suggestions for Authors

The authors studied a prediction method for viral immunogenicity using advanced machine learning methods. Overall, the study design and methodologies, and description were good. However, two problems were found.

1. Please provide subheadings to the Results section and arrange them.

2. There was no comparisons with between previous reports and this study  in Introduction and Discussion. Moreover, Furthermore, there is no actual description of the predictions regarding genetic mutations, amino acid substitutions, and antigenic changes in Results and Discussion.

Round 2

Reviewer 1 Report

Comments and Suggestions for Authors

I am generally satisfied with the additions made by the authors to the manuscript.

However, one of the phrases present in the authors' responses to my comments confused me: The manuscript containing this enriched dataset has been submitted to another journal for publication.

This statement requires additional verification of the issue of publication of the same results in the current article and the article mentioned by the authors, until clarification of which this article cannot be considered for publication.

Hoping that there are no problems with plagiarism here, I propose for now a minor revision so that the authors clarify the issue.

Comments on the Quality of English Language

      No big problems

Reviewer 2 Report

Comments and Suggestions for Authors

The authors well addressed for all comments by the Reviewer. Thus, the revised manuscript was acceptable for publication in this journal.

Author Response

We express our gratitude to the reviewer for accepting our manuscript for publication.